# Anomalous Salt Dependence Reveals an Interplay of Attractive and Repulsive Electrostatic Interactions in α-synuclein Fibril Formation

Asymmetric charge distribution; electrostatic interactions; secondary nucleation; protein aggregation; self-assembly; salt screening

**Author for correspondence:**
Sara Linse,
E-mail: sara.linse@biochemistry.lu.se

Ricardo Gaspar[1,2], Mikael Lund[3,4] (ID), Emma Sparr[1] and Sara Linse[2] (ID)

[1]Department of Physical Chemistry, Lund University, Lund, Sweden; [2]Department of Biochemistry and Structural Biology, Lund University, Lund, Sweden; [3]Department of Theoretical Chemistry, Lund University, Lund, Sweden and [4]Lund Institute of Advanced Neutron and X-ray Science, Lund, Sweden

## Abstract

α-Synuclein (α-syn) is an intrinsically disordered protein with a highly asymmetric charge distribution, whose aggregation is linked to Parkinson's disease. The effect of ionic strength was investigated at mildly acidic pH (5.5) in the presence of catalytic surfaces in the form of α-syn seeds or anionic lipid vesicles using thioflavin T fluorescence measurements. Similar trends were observed with both surfaces: increasing ionic strength reduced the rate of α-syn aggregation although the surfaces as well as α-syn have a net negative charge at pH 5.5. This anomalous salt dependence implies that short-range attractive electrostatic interactions are critical for secondary nucleation as well as heterogeneous primary nucleation. Such interactions were confirmed in Monte Carlo simulations of α-syn monomers interacting with surface-grafted C-terminal tails, and found to be weakened in the presence of salt. Thus, nucleation of α-syn aggregation depends critically on an attractive electrostatic component that is screened by salt to the extent that it outweighs the screening of the long-range repulsion between negatively charged monomers and negative surfaces. Interactions between the positively charged N-termini of α-syn monomers on the one hand, and the negatively C-termini of α-syn on fibrils or vesicles surfaces on the other hand, are thus critical for nucleation.

## Introduction

Accumulation of the protein α-synuclein (α-syn) in the form of amyloid deposits coincides with the loss of dopaminergic neurons in Parkinson's disease (PD; Spillantini *et al.*, 1997; Breydo *et al.*, 2012). A key feature of α-syn is its highly asymmetric charge distribution with an abundance of negatively charged residues in the C-terminal region and the positive charges located mainly in the N-terminal region (Fig. 1). A large fraction of the 140 amino acid residues are hydrophobic and these are particularly abundant among the first 100 residues. The central region of α-syn (residues 61–95) carries zero net charge.

In its monomeric state, α-syn is an intrinsically disordered protein, while it folds to distinctly different structures upon binding to anionic lipid membranes or upon fibril formation. In the membrane-bound state, the protein is embedded in the head group and upper acyl layer of the phospholipid membrane (Pfefferkorn *et al.*, 2012; Hellstrand *et al.*, 2013a) with the first ca. 100 residues taking up an α-helical conformation (Davidson *et al.*, 1998; Eliezer *et al.*, 2001; Fusco *et al.*, 2014; Galvagnion *et al.*, 2015). In the fibrillar state, several of the hydrophobic side-chains of the central region are buried in the core of the amyloid fibril, as revealed by recent structural investigations (Der-Sarkissian *et al.*, 2003; Tuttle *et al.*, 2016; Guerro-Ferreiera *et al.*, 2018; Li *et al.*, 2018a; 2018b). These fibrils display a repetitive packing of the protein monomers with β-strands running perpendicular to the fibril axis, thus forming extended β-sheets along the length of the fibril. The α-syn fibrils thus share many of the generally observed features of amyloid fibrils (Dobson, 1999; Eisenberg and Sawaya, 2017). However, with a large number of buried residues, the monomer organization in each plane of the fibril is more complex than in the high-resolution structures of model peptide fibrils (Eisenberg and Sawaya, 2017), and as a consequence, α-syn fibrils rather resemble amyloid fibrils formed from long peptides and proteins such as those of the amyloid β peptide and protein tau (Colvin *et al.*, 2016; Fitzpatrick *et al.*, 2017). Only residues 37–99 are seen in the ordered part of the fibril, thus the fibril is likely decorated by polymer-brush like structures of the slightly positive N-terminal and the highly negatively charged C-terminal tail. Both sequence and solution conditions may contribute to the short-range arrangement, to the packing of protofilaments into fibrils, as well as, to the higher order association of fibrils into bundles, networks or clumps (Marshall *et al.*, 2011; Fujiwara *et al.*, 2019; Pogostin *et al.*, 2019). Thus, depending on the conditions, α-syn fibrils are observed with more than one type of local packing, which is translated to different morphologies at longer length

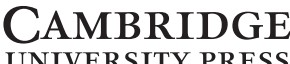

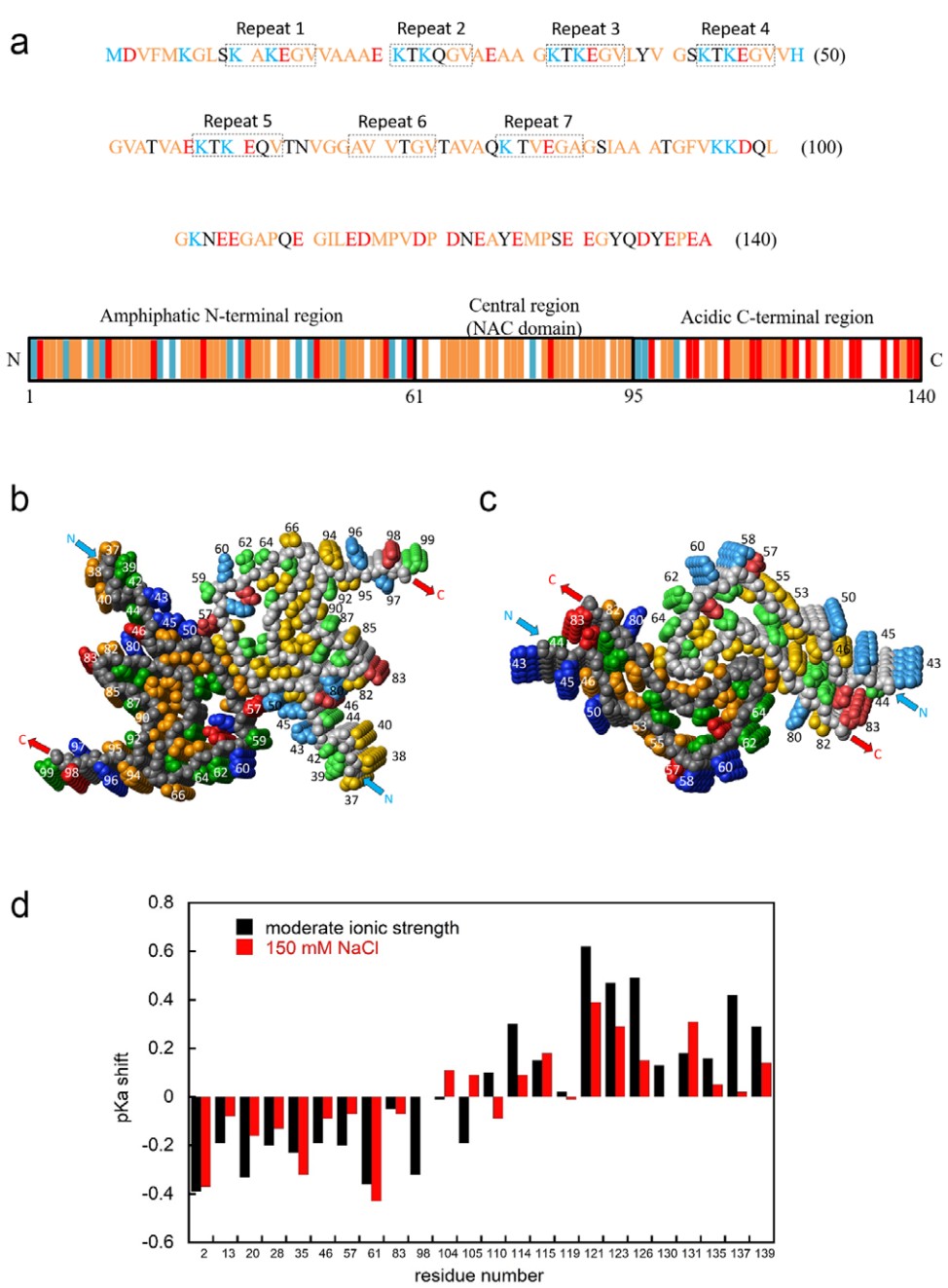

**Fig. 1.** (*a*) Amino acid sequence of α-syn. Orange letters indicate hydrophobic residues, and black letters indicate uncharged polar residues. Red and blue indicate acidic and basic residues, respectively. The consensus sequences are marked with a dotted line box. The colored panel below the sequence highlights the asymmetric distribution of charged residues in α-syn with the classical region notation above the cartoon. NAC stands for non-amyloid component, to distinguish it from β-amyloid, but indeed forms the core of α-syn amyloid fibrils. (*b*) Six planes of α-syn fibrils in one conformation reported by several investigators (6a6b.pdb; Li *et al.*, 2018*b*). (*c*) Five planes of α-syn fibrils in another polymorph (6cu8.pdb; Li *et al.*, 2018*a*) with surface-exposed residues numbered and in each plane there are two monomers, one shown in lighter colors and one in darker colors with hydrophobic, acidic, basic and titrating residues indicated. The connections to the termini are indicated by arrows marked N and C, respectively. (*d*) Histogram showing the shift in p$K_a$ values of the 25 acidic groups reported for 0.25 mM α-syn in 20 mM sodium phosphate and in 20 mM sodium phosphate with 150 mM NaCl (Croke *et al.*, 2011).

scales (Heise *et al.*, 2005; Li *et al.*, 2018*a*). Only structures that accommodate the side-chains without steric hindrance are observed, and the structures that do form, typically have neatly packed side-chains in a repetitive pattern (Eisenberg and Sawaya, 2017).

The assembly of α-syn monomers from a supersaturated solution to fibrillar aggregates follows a mechanism that involves at least three microscopic steps: primary nucleation of monomers, elongation of existing fibrils by monomer addition and secondary nucleation of α-syn monomers on the surface of existing fibrils. The rates of all three steps are sensitive to intrinsic (sequence) (Rivers *et al.*, 2007) and extrinsic (i.e. pH, temperature, and surfaces) (Munishkina *et al.*, 2004; Fink, 2006; Buell *et al.*, 2014) factors, the study of which can further deepen our knowledge of mechanistic details of aberrant protein misfolding (Wu *et al.*, 2009). In pure buffer, monomers of α-syn remain disordered for prolonged time and aggregation may not be detected for up to several days as aresult of a very low rate of primary nucleation.

However, heterogeneous primary nucleation leads to catalysis of aggregation. This is relevant to all conditions when the proteins are present together with biological or foreign surfaces in the form of, for example, lipid membranes (Lee *et al.*, 2002*a*; 2002*b*; Galvagnion *et al.*, 2015) and nanoparticles (Vácha *et al.*, 2014). After the first primary nucleation and elongation events, the aggregation process is controlled either by elongation or by secondary nucleation of monomers on existing fibril surfaces, depending on pH conditions. Elongation of fibrils by monomer addition is faster than nucleation and relatively insensitive to pH (Buell *et al.*, 2014; Gaspar *et al.*, 2017). For mildly acidic conditions, the process of secondary nucleation of monomers on existing fibril surfaces is the dominant microscopic event of the aggregation mechanism. Under these conditions, the aggregation of α-syn is rapid due to this autocatalytic amyloid amplification process (Gaspar *et al.*, 2017; Törnquist *et al.*, 2018). Thus, pH changes serve as a mechanistic switch, in which secondary nucleation of α-syn monomers on the surface of existing fibrils is undetectably slow above pH 6.0 (Buell *et al.*, 2014). Note also that membrane-induced aggregation is more efficient at mildly acidic conditions compared to neutral pH, where systematic studies have revealed that only certain lipid systems, such as, lipids with short (C12–C14) saturated acyl chains and PS head groups trigger α-syn aggregation (Galvagnion *et al.*, 2016; Gaspar *et al.*, 2018). At neutral pH with added salt, both surface catalysed aggregation processes, secondary nucleation and heterogeneous primary nucleation were shown to be inhibited (Buell *et al.*, 2014; Galvagnion *et al.*, 2015).

Protein–protein interactions are relevant to protein function and solution behaviour as well as nucleation processes. The effective interactions originate from a complex combination of intermolecular interactions between protein residues (Schreiber and Fersht, 1995; Jones and Thornton, 1996; Lo Conte *et al.*, 1999; McManus *et al.*, 2007; Li *et al.*, 2014). A detailed understanding of the free energy of complex formation requires knowledge of protein structures, which reveal how the amino acids are organized at the protein surface and interface (Lo Conte *et al.*, 1999). Surface heterogeneity is a common feature of proteins, that is anisotropic patterns of neutral, charged and hydrophobic patches, which influence both affinity and specificity of inter-molecular interactions and their orientational dependency (Harris, 1950; Perutz, 1978; Schreiber and Fersht, 1995; Jones and Thornton, 1996; Lo Conte *et al.*, 1999; McManus *et al.*, 2007; Li *et al.*, 2014; Li *et al.*, 2016*a*; Tesei *et al.*, 2017). The uneven distribution of ionizable residues in α-syn implies that their p$K_a$ values, and thereby the charge profile, strongly depend on solution conditions, such as pH and ionic strength (Fig. 1*d*; Wu *et al.*, 2009; Croke *et al.*, 2011; Buell *et al.*, 2014). Although burial of hydrophobic side chains is the main driving force for amyloid formation, the rates and equilibrium parameters are modulated by electrostatic interactions and may be tuned by changes in ionic strength or charge substitutions (Zurdo *et al.*, 2001; Betts *et al.*, 2008; Juárez *et al.*, 2009; Muzzaffar and Ahmad, 2011; Portillo *et al.*, 2012; Buell *et al.*, 2013; 2014; Abelein *et al.*, 2016; Meisl *et al.*, 2017, Ranjan and Kumar, 2017; Yang *et al.*, 2018).

In the present study, the kinetics of α-syn amyloid formation was monitored in the presence of catalytic surfaces in the form of preformed α-syn seeds or anionic lipid vesicles. The goal was to investigate the role of electrostatic interactions in secondary nucleation and in heterogeneous primary nucleation at vesicle surfaces. To enable this, our studies were conducted at mildly acidic conditions (pH 5.5) with systematic variation of the salt concentration covering conditions relevant for extracellular and intracellular

spaces (Wennerström *et al.*, in press) as well as more extreme salt concentrations (high and low). Interestingly, although these different catalytic surfaces enhance aggregation by triggering distinct nucleation processes, the effect of ionic strength was shown to be similar in both cases leading to a progressive retardation of aggregation with increasing salt concentration. The results imply that charge interactions are crucial for aggregation to occur at a detectable rate. The experiments were complemented by Monte Carlo (MC) simulations of monomers interacting with a surface decorated by α-syn C-terminal tails as a function of salt concentration.

## Results

### Salt dependence of seed-catalysed α-syn amyloid fibril formation

The aggregation of α-syn was monitored using thioflavin-T (ThT) fluorescence in non-binding PEGylated plates in the presence of catalytic preformed α-syn seeds, thereby the contribution from primary nucleation is negligible (Gaspar *et al.*, 2017). To further investigate this autocatalytic process, our studies were conducted at mildly acidic pH. It is important to highlight that the pH dependency of α-syn aggregation in 10 mM phosphate buffer as previously reported (Buell *et al.*, 2014) was replicated with our experimental 10 mM MES buffer (Fig. S1), implying that it is not strongly affected by changing the buffer composition, as long as, the ionic strength is not varied substantially. Here, MES was chosen to provide a lower ionic strength baseline compared to phosphate.

To investigate the role of electrostatic interactions on fibril formation, an initial kinetic aggregation experiment was performed in 10 mM MES buffer pH 5.5 + 140 mM NaCl, in the presence of 0.003, 0.03, 0.3 or 3 μM seeds, with monomer concentrations in the range of 0.5–30 μM. No aggregation was detected during the time frame of the experiment (20 h) (Fig. S2). This motivated systematic studies of the effect of ionic strength to elucidate the importance of charge interactions.

The aggregation kinetics at a fixed α-syn monomer concentration (5 μM) at two different seed concentrations (0.005 and 0.05 μM) was therefore followed for multiple samples with NaCl concentrations ranging from 0 to 1,200 mM (Fig. 2). First, in the absence of added salt (black traces, Fig. 2) the overall aggregation takes place within a few hours, in agreement with earlier findings (Buell *et al.*, 2014; Gaspar *et al.*, 2017). By definition, secondary nucleation leads to an acceleration of the aggregation at early time points, which is manifested through observable ThT sigmoidal traces that have concave shapes (Buell *et al.*, 2014; Gaspar *et al.*, 2017). Addition of salt progressively retards the aggregation with visible changes in the slope of the observable macroscopic traces (Fig. 2). At 200 mM NaCl or more, fibril formation is completely inhibited for the 64 h duration of the experiment for both seed concentrations investigated (Fig. 2).

### Kinetic analysis

The experimental data up to 120 mM NaCl was analysed by fitting an aggregation kinetic model that includes secondary nucleation and elongation (Fig. 2*d*,*e*; Meisl *et al.*, 2016) to the data to gain information about the effect of ionic strength on the rate constants for nucleation and elongation. As the reaction occurs in the presence of seeds, the contribution from primary nucleation ($k_n$) is overruled and therefore its rate constant becomes negligible. The resulting fits provide estimates of the product of the rate constants

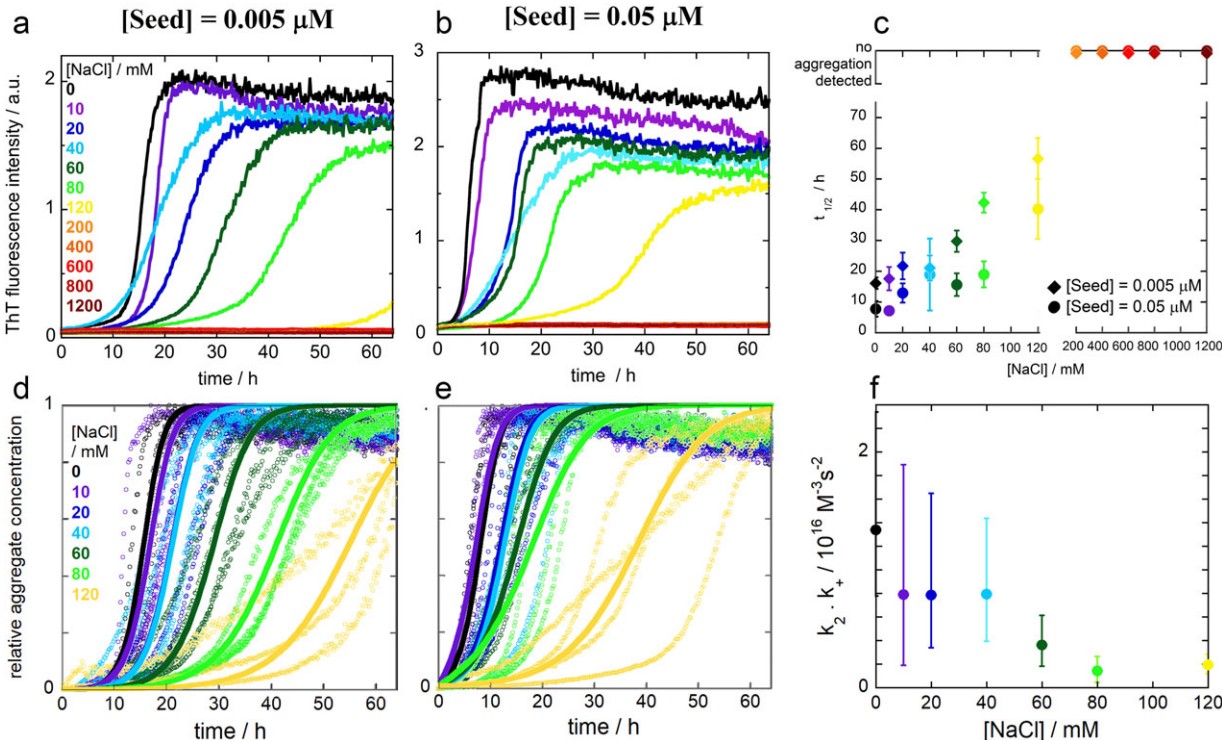

**Fig. 2.** Salt dependence of the α-syn aggregation kinetics. Aggregation kinetics monitored using thioflavin-T (ThT) fluorescence starting from 5 μM α-syn monomer in the presence of two different seed concentrations, (*a*) 0.005 and (*b*) 0.05 μM, were monitored at salt concentrations ranging from 0 to 1,200 mM NaCl in 10 mM MES buffer pH 5.5 at 37°C and under quiescent conditions with color codes shown in panel (*a*). The figures show the median traces of at least three experimental repeats. (*c*) $t_{1/2}$, the time at which the ThT fluorescence has reached 50% of the total fluorescence amplitude, as a function of salt concentration for both seed concentrations. Normalized aggregation kinetic traces for 5 μM α-syn monomer in the presence of (*d*) 0.005 μM and (*e*) 0.05 μM seed at salt concentrations ranging from 0 to 120 mM NaCl. The figures show the experimental repeats dotted with the fits as solid lines. (*f*) Fitted values of the rate constant product $k_2 k_+$ as a function of salt concentration.

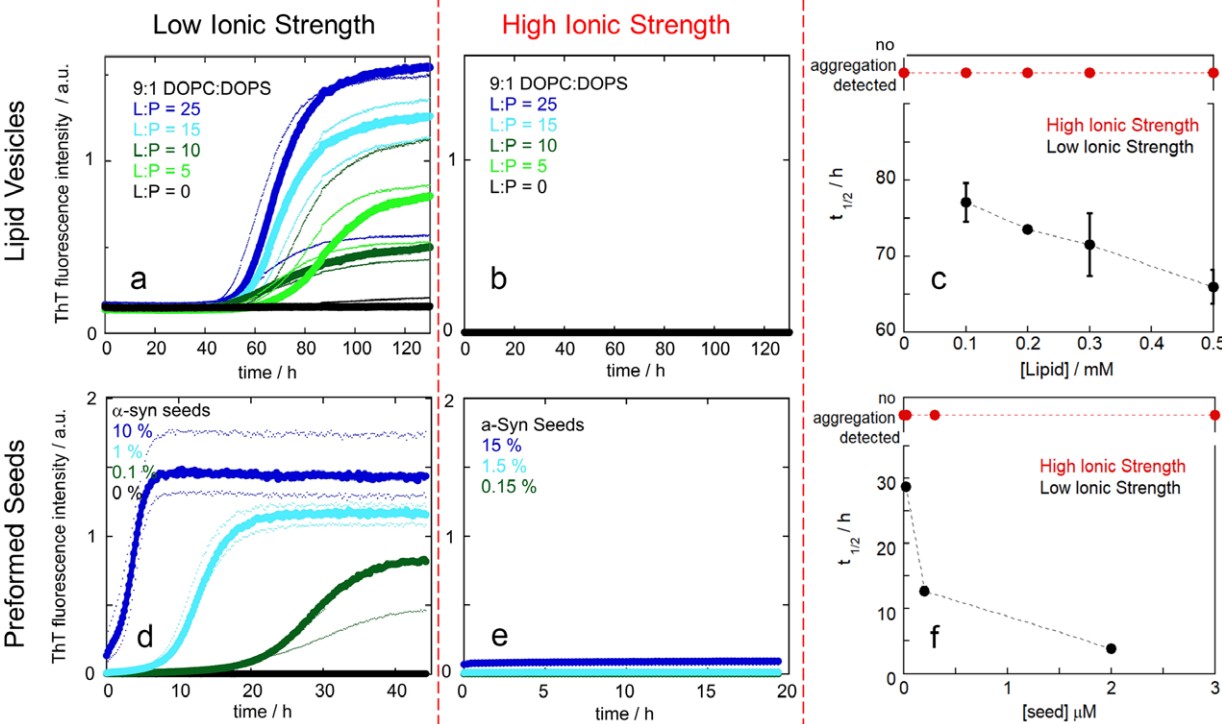

**Fig. 3.** The effect of ionic strength on aggregation kinetics of α-syn in the presence of different catalytic surfaces. Aggregation kinetics starting from 20 μM α-syn monomer was monitored by thioflavin-T (ThT) in the presence of two different catalytic surfaces, seeds and anionic lipid vesicles, in 10 mM MES buffer pH 5.5 at 37°C under quiescent conditions with and without 140 mM added NaCl. Thick lines represent the median traces of at least three experimental repeats (dotted).

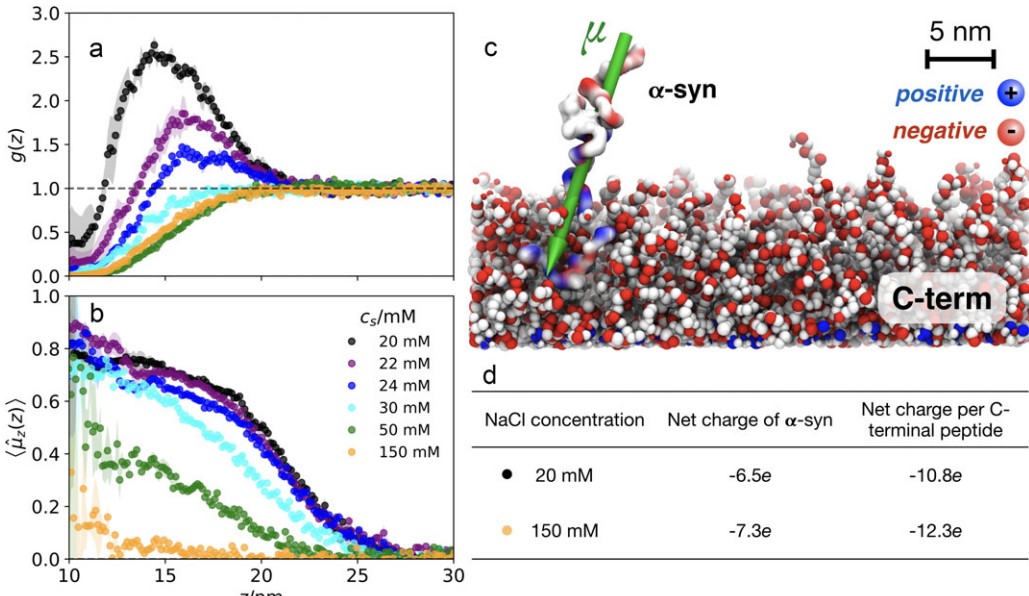

**Fig. 4.** Results of Monte Carlo (MC) simulations. (*a*) Mass centre density of α-syn relative to bulk as a function of distance, *z*, to the grafting surface of C-term. (*b*) The corresponding degree of dipole moment orientation. The simulations were carried out at pH 5.5 for a range of different salt concentrations, $c_s$, and represent thermal averages over all α-syn and C-term configurations and protonation states. The translucent regions show the standard deviation as estimated from two MC independent runs. (*c*) Configuration rendered from one simulation of α-syn (surface representation) interacting with a grafted layer of C-term (red/white/blue beads). The green arrow represents the molecular dipole moment vector, $\mu$, of α-syn in one typical conformation obtained during the simulations. The simulation is carried out at 20 mM salt and pH 5.5 using a constant pH ensemble. (*d*) Calculated average net charge of α-syn and grafted C-terminal peptides, from simulations at low and high salt concentration at pH 5.5.

of elongation ($k_+$) and secondary nucleation ($k_2$). The obtained values of the product $k_2 k_+$ are plotted as a function of salt concentration in Fig. 2*f*. We find that $k_2 k_+$ decreases progressively with increasing salt concentration, implying a strong ionic strength dependency.

### Salt dependence of lipid membrane-induced α-syn amyloid fibril formation

Aggregation kinetic experiments were performed in systems where the monomeric protein was added to small unilamellar vesicles composed of mixtures of phospholipids, [1,2-dioleoyly-*sn*-glycero-3-phosphocholine (DOPC):1,2-dioleoyl-*sn*-glycero-3-phospho-L-serine sodium salt (DOPS), molar ratio 9:1) at varying salt concentrations (0 or 140 mM NaCl) and lipid-to-protein ratios (0–25; Fig. 3*a,b*). These lipids were chosen for their biological relevance as both PC and PS are abundant lipid components of cell membranes. At the present pH of 5.5, the PS headgroups are expected to be predominantly in an anionic form (charge −1) (Tokutomi *et al.*, 1980; Tsui *et al.*, 1986), whereas PC headgroups are expected to be uncharged. All experiments were performed at conditions when the vesicles surface is saturated with protein, which has been shown to be a requirement for aggregation to occur (Galvagnion et al., 2015).

At low ionic strength, lipid vesicles trigger aggregation of α-syn. This is consistent with previous reports for different lipid anionic membranes at similar salt concentrations (Galvagnion *et al.*, 2016; Gaspar *et al.*, 2018). However, when the salt concentration is increased to 140 mM NaCl, no aggregation is detected within the time frame of the experiment. Thus, we conclude that α-syn fibril formation has similar salt dependency in the presence of different types of catalytic surfaces, including α-syn fibril seeds and anionic lipid vesicles (Fig. 3*c,f*).

Previous studies have shown that the adsorption of α-syn to DOPC:DOPS lipid bilayers strongly depends on the solution

conditions, forming clearly thinner protein layers at the higher salt concentrations (Hellstrand *et al.*, 2013*a*). The same behaviour was confirmed for the present lipid mixture (Fig. S3).

### Molecular simulations

To obtain mechanistic insight on *how* an α-syn monomer interacts with a C-termini grafted surface, molecular MC simulations were performed. We use a coarse-grained peptide model, previously developed to study electrostatic interactions of intrinsically disordered proteins in electrolyte solutions (Cragnell *et al.*, 2018), see details in the Materials and Methods section. The simulated system is composed of one full-length α-syn monomer and a surface with α-syn C-terminal tails (residues 101–140) grafted at a density of one tail per 1,200 Å$^2$. This number was based on available information on fibril structure and lipid to protein ratio at saturation (Galvagnion *et al.*, 2016; Li *et al.*, 2018*a*; 2018*b*; see Methods in Supplementary Materials). From these simulations we extract the α-syn mass centre distribution function with respect to the surface ($g(z)$, Fig. 4*a*). At high salt concentration, $g(z)$ is smaller than one, that is the protein is repelled from the surface due to the reduced peptide conformational space upon approaching the interface. However, at low ionic strength, we observe a pronounced peak in $g(z)$, indicating attractive inter-molecular interactions, despite that the overall charge of the surface-associated protein molecules, in particular their exposed C-termini, and the free α-syn monomers both are net negatively charged. To understand this seemingly counter-intuitive observation, we track the average dipole orientation of α-syn as a function of the mass centre separation to the surface (Fig. 4*b*). We find that the full-length protein is strongly oriented with the positively charged part embedded in the negatively charged C-terminal tail layer on the surface.

Decreasing the salt concentration, we observe stronger and stronger alignment, meaning that the positive end of α-syn points

towards the negatively charged surface (Fig. 4*c*). The theoretical maximum alignment is 1.0 and for the case of 20 mM salt, the degree of alignment thus approaches 80%. At high salt concentration, the electrostatic repulsion between the surface and the negatively charged end of the monomer is mostly screened and on average, no particular dipolar orientation is taken.

The MC simulations were performed at constant pH where the charge states of ionizable amino acids fluctuate according to the local environment. Both α-syn and the grafted C-termini are negatively charged at pH 5.5 (Fig. 4*d*), yet a net attraction is observed at low salt concentrations (Fig. 4*a*). At elevated salt concentrations, the absolute peptide charges increase since the local electric potential felt by any ionizable site is reduced. That is, at high salt all protonation states tend towards the value dictated by the intrinsic p$K_a$ values.

## Discussion

### Counter-intuitive salt dependence

To understand the importance of charge–charge interactions in α-syn aggregation, the electrostatic component was here modulated by varying the ionic strength for mildly acidic pH conditions. With an iso-electric point around 4.7, α-syn is net negatively charged at pH 5.5. Intriguingly, the aggregation of α-syn is retarded upon addition of 140 mM NaCl and no aggregation was observed even in the presence of anionic vesicles or seeds at high concentrations. These two catalytic surfaces, anionic lipid vesicles and α-syn seeds, influence the aggregation mechanism differently by acting on primary or secondary nucleation, respectively. Yet the effect is in each case highly dependent on the ionic strength.

In previous studies of another net negative amyloid-forming peptide, Aβ, screening of electrostatic repulsion by salt was found to increase the overall aggregation rate. In particular, the rate of secondary nucleation was enhanced upon screening of the repulsion between the negatively charged monomers and between monomers and the negatively charged fibril surface (Abelein *et al.*, 2016; Meisl *et al.*, 2017). This result can be rationalized by salt screening of the repulsive electrostatic interaction between two like-charged objects. The presented results for α-syn are opposite in that salt reduces the aggregation rate despite that both monomers and fibrils of α-syn are net negatively charged. This suggests that the aggregation mechanism contains an attractive electrostatic component, since screening of the long-range repulsion between negatively charged monomer and seed species does not lead to faster aggregation kinetics (Bratko *et al.*, 2002). An attractive electrostatic interaction was indeed found in the MC simulations of α-syn monomers in solution next to a surface with grafted α-syn C-termini. The simulations identified critical attractive interactions between the positively charged N-terminus of the monomer in solution with the negatively charged C-termini on the surface, yielding a preferred orientation (Fig. 4*c*), which were screened by salt (Fig. 4*b*). Despite its simplicity, the peptide model used in the MC simulations have previously been shown to agree well with small angle X-ray scattering experiments for charged IDP's in salt solutions (Cragnell *et al.*, 2018). Arguably this success can be attributed to the fact that the model well describes electrostatic interactions for systems where these dominate. As our experimental results show, electrostatics indeed seem governing in the present system, and we hence expect the simulations to provide a fair, phenomenological view of the principal physics.

### Proposed mechanism

We propose a plausible scenario that would explain the observed anomalous salt dependence of α-syn aggregation in the presence of anionic membranes or seeds, based on the anisotropic α-syn charge distribution (Fig. 5). If interactions between the positively charged N-terminal region of α-syn monomers and the negatively charged C-terminal tails of α-syn, which may be decorating the fibrils (Tuttle *et al.*, 2016; Guerro-Ferreiera *et al.*, 2018; Li *et al.*, 2018*a*; 2018*b*), are critical for nucleation on the surface of the fibrils, then this attractive electrostatic component will be diminished upon screening by salt (Figs 4*b* and 5*b*). Indeed, the results of a recent nuclear magnetic resonance (NMR) study reveal electrostatic interactions between charged regions in the N- and C-termini, which are screened by salt (Doherty *et al.*, 2020). Our results are thus compatible with secondary nucleation relying on an electrostatic attraction between the positively charged N-terminus of α-syn monomers and negatively charged C-terminal tails decorating the pre-formed fibrils. Other scenarios are also plausible, such as the collapse of the C-terminal tail, or lateral association of mature fibrils which could reduce the surface area available for secondary nucleation. Likewise, if interactions between the positively charged N-terminal regions of α-syn monomers and the negatively charged C-terminal tails of α-syn, which may be decorating vesicles with adsorbed α-syn monomers (Ferreon *et al.*, 2008; Gaspar *et al.*, 2018), are critical for heterogeneous primary nucleation on the vesicle surface, then this attractive electrostatic component will be diminished upon screening by salt (Fig. 5*a*,*b*). Our results are thus compatible with heterogeneous primary nucleation relying on an electrostatic attraction between the positively charged N-terminus of α-syn monomers and negatively charged C-terminal tails decorating the phospholipid membrane. This scenario is well in line with the noted requirement of an α-syn concentration above what gives full coverage of the membrane surface for aggregation to be observed in the presence of vesicles (Fusco *et al.*, 2014).

### Anisotropic charge distribution may yield anomalous salt dependence

There is ample evidence that protein charges are important for solubility and function. However, the highly asymmetric charge distribution of α-syn (Fig. 1) may be linked both to its function and dysfunction. Uneven charge distributions are common in proteins and have for other systems been linked to protein function (Klapper *et al.*, 1986; Sines *et al.*, 1990; Antosiewicz *et al.*, 1996; Chirgadze and Larionova, 1999; Kesvatera *et al.*, 2001), as well as, low-affinity protein–protein interactions due to so-called 'patchy attraction' (Li *et al.*, 2014, Li *et al.*, 2016*a*; Li *et al.*, 2016*b*). Similar to protein folding, high affinity protein–protein interactions and protein self-assembly processes are governed by interactions involving hydrophobic residues. However, the free energy of folding or binding is modulated by electrostatic interactions, and an anisotropic charge distribution may play a significant role (Chirgadze and Larionova, 1999; Parker *et al.*, 2012). For example, uneven charge distributions may act to increase binding rates through electrostatic steering of substrates to active sites as reported for, for example, superoxide dismutase (Klapper *et al.*, 1986; Sines *et al.*, 1990) and acetylcholinesterase (Antosiewicz *et al.*, 1996). While proteins are destabilized by a high net charge (Akke and Forsén, 1990; Lindman *et al.*, 2006*a*; Matousek *et al.*, 2007), this effect can be attenuated by an uneven charge distribution between subdomains (Linse *et al.*, 2000). Short peptides may self-assemble in an arrangement that

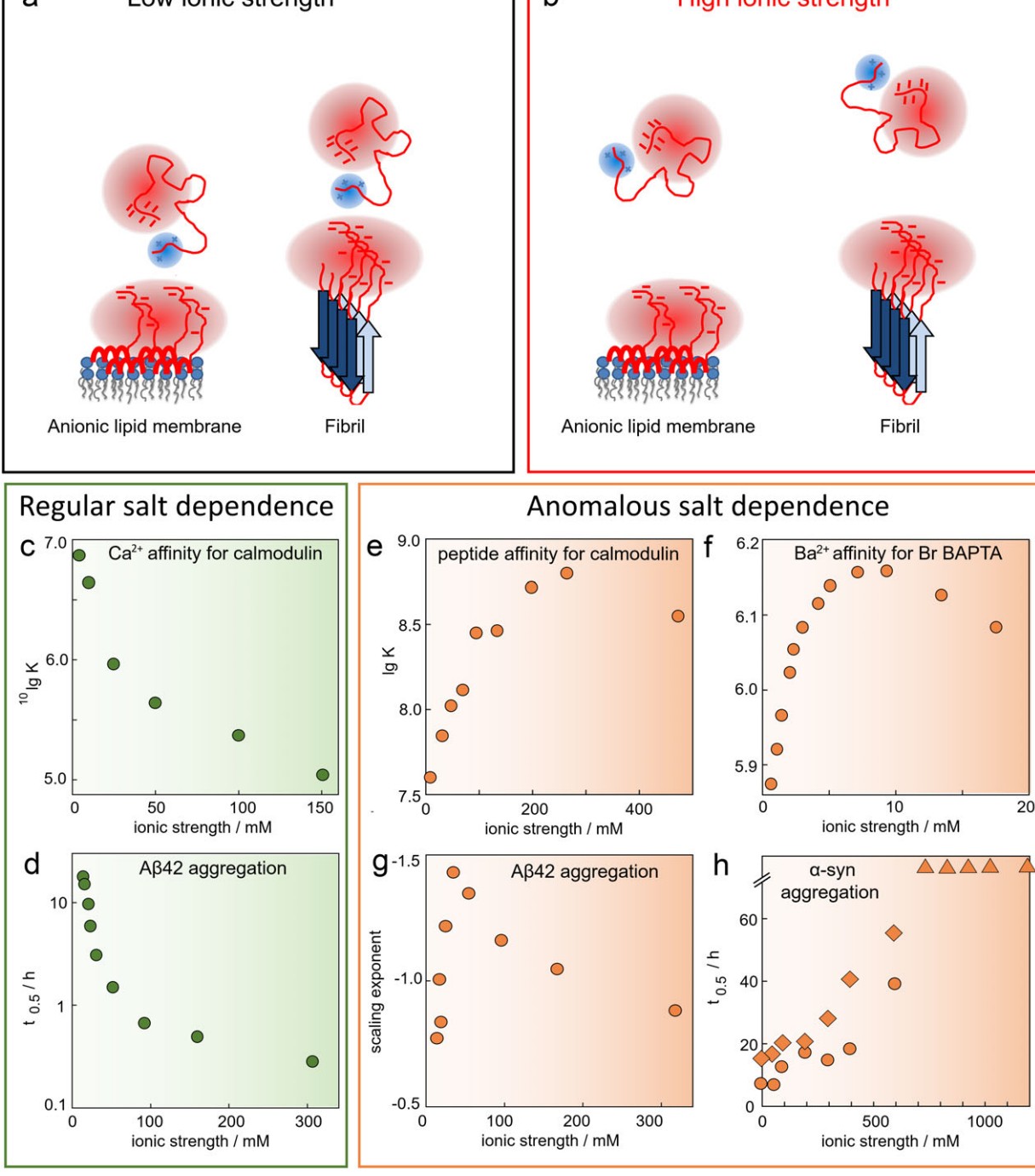

**Fig. 5.** Asymmetric charge distribution can yield anomalous salt dependence. (*a* and *b*) Cartoon illustrating what incoming α-syn monomers in solution sense. In the case of membrane-bound α-syn or α-syn in fibrils, it has been suggested that α-syn negative C-terminal tails are sticking out into solution decorating the vesicles and fibrils. The positive N-terminus of α-syn leads to preferential orientation of the monomer relative to the negatively charged surfaces due to attractive electrostatic interactions dominating at low ionic strength (*a*). At high ionic strength (*b*) the orientational preference is reduced. Examples of systems showing regular (*c* and *d*) and anomalous (*e–h*) salt dependence. The data are from published work in panels *d* and *g* (Meisl *et al.*, 2017), *e* (André *et al.*, 2006), *f* (da Silva *et al.*, 2005), *c* (Linse *et al.*, 1991) and from this work in panel *h*.

minimizes the electrostatic repulsion between neighbouring chains in the fibril (Diaz-Espinoza *et al.*, 2017).

Electrostatic interactions are screened by salts, macromolecular ions and their counter-ions. Typically, repulsive interactions between opposite charges are weakened (Fig. 5*c,d*) and attractive interactions between like charges are also weakened, but in complex systems with anisotropic charge distributions, the overall

observable effect of electrostatic screening may be anomalous (Fig. 5*e–h*). One example of this is found for the interaction between $Ca_4$-calmodulin (net charge −15) and a target peptide from smooth muscle myosin light chain kinase (charge +8*e*), which is stronger in the presence of 100 mM NaCl compared to low salt, implying that the net electrostatic interactions disfavors complex formation (Fig. 5*e*; André *et al.*, 2006). The two-domain

structure of calmodulin, with a significant reduction of the domain–domain separation in the complex (Hellstrand *et al.*, 2013*b*), gives a strong repulsive electrostatic component that over-rides the electrostatic attraction of the target peptide, and screening therefore leads to higher affinity in the presence of salt. Another example of anomalous salt dependence includes metal ion binding to small chelators in the presence of silica nanoparticles (Fig. 5*f*; da Silva *et al.*, 2005). Here salt at moderate concentrations screens the effect of the silica macro-ions on the chelator-metal ion interaction, leading to an increase in affinity. Salt at high concentration produces an effect in the opposite direction leading to a decrease in affinity due to the dominant screening of the chelator-metal ion interaction. The salt dependence thus becomes biphasic in this system. Another case is the aggregation of Aβ42, for which a biphasic salt dependence is observed for the scaling exponent of the half time dependence on monomer concentration (Fig. 5*g*; Meisl *et al.*, 2017).

Abnormal salt effects on intermolecular interactions between like-charged biomolecules with uneven charge distributions have previously been observed for antibodies (Roberts *et al.*, 2014), lactoferrin (Li *et al.*, 2016*a*), and arginine-rich peptides (Tesei *et al.*, 2017). A shared feature of these systems is that the electrostatic repulsion due to the equally signed net-charges contends with an opposing electrostatic attraction stemming from interactions between charged surface patches. This balance is sensitive to subtle changes in salt concentration and leads to a nonmonotonic salt effect on intermolecular interactions. The mechanism is compatible with the experimental results presented here and is directly observed in the presented MC simulations.

### $pK_a$ value shifts in charged proteins

Uneven charge distributions are manifested in site-specific $pK_a$-values of ionizable residues in proteins. Groups that titrate in an overall negatively charged environment have up-shifted $pK_a$ values, that is high proton affinity, while groups that titrate in an overall positively charged environment have down-shifted $pK_a$ values. This is indeed the case for α-syn where the $pK_a$ values of the 25 acidic groups and the single histidine side-chain have been reported for 0.25 mM α-syn in 20 mM sodium phosphate (Fig. 1; Croke *et al.*, 2011). Another contributing factor is that the $pK_a$ values of ionizable residues may be affected by added salt (Abe *et al.*, 1995; Lee *et al.*, 2002*a*; 2002*b*). Increasing the ionic strength shifts the $pK_a$ values closer to model compound values, resulting in an increase in the overall net negative charge (Fig. 4*d*). Less shifted $pK_a$ values at higher salt concentration (150 mM NaCl in 20 mM sodium phosphate for 0.25 mM α-syn, Croke *et al.*, 2011) agrees with previous reports for histidine side-chains in *Staphylococcus* nuclease (Lee *et al.*, 2002*a*; 2002*b*), as well as, acidic residues in lysozyme (Abe *et al.*, 1995) and PGB1 (Lindman *et al.*, 2006*b*; 2007). In the case of calbindin D$_{9k}$, it was found that increasing the protein concentration has a similar effect of increased electrostatic screening with more prominent $pK_a$ shifts of basic residues observed at low protein concentration (Kesvatera *et al.*, 1996).

The $pK_a$ shifts reported for α-syn are smaller than observed for some folded proteins for which up-shifts of 1.5–2.0 units have been reported (Kesvatera *et al.*, 1999; 2001; Castañeda *et al.*, 2009). In part, this may be due to the relatively high ionic strength of the buffer used in the $pK_a$-value measurements for α-syn (Croke *et al.*, 2011). Still, it is clear that the $pK_a$ values of the acidic residues, which are highly concentrated in the C-terminal tail of α-syn, are less up-shifted at elevated salt concentrations, making this part of

the protein more negatively charged with increasing ionic strength (Fig. 4*d*). This could lead to greater repulsion between α-syn monomer and fibril surfaces, but is compensated by increased electrostatic screening. Thus, we need an alternative explanation of the anomalous salt dependence, which we suggest arises from screening of attractive interactions between the incoming monomers and the negative tails decorating the membrane surfaces and fibrils. In line with this interpretation, a recent study of C-terminally truncated α-syn variants show reduced salt dependence of secondary nucleation (van der Wateren *et al.*, 2018).

### Concluding remark

Multiple studies have demonstrated that the aggregation properties of amyloidogenic peptides are strongly affected by the environment, where electrostatics play an important role in defining both aggregation kinetics and final morphology of the aggregated species (Portillo *et al.*, 2012). The current study focuses on the biophysical properties of α-syn aggregation and brings novel and essential knowledge in terms of molecular mechanisms that may occur *in vivo*. Our results provide insights into the importance of electrostatic interactions in amyloid formation, and quite evidently shows the strong dependency of α-syn aggregation on solution conditions, which also varies *in vivo* between different intracellular environments. In particular, we find that the anomalous salt dependence of α-syn aggregation cannot be explained based on the peptide net charge, but requires consideration of its asymmetric charge distribution resulting in an attractive electrostatic component between the positively charged N-terminal region of incoming monomers and the negatively charged C-terminal tails that decorate fibrils or vesicles with adsorbed protein.

### Materials and methods

#### α-Syn expression and purification

Human α-syn was expressed in *Escherichia coli* from a Pet3a plasmid containing a synthetic gene with *E. coli*-optimized codons (Purchased from Genscript). α-Syn was purified using heat treatment, ion exchange and gel filtration chromatography, as described (Grey *et al.*, 2011).

For all experiments, α-syn was purified by size exclusion in the desired experimental buffer to assure purely monomeric form of the peptide.

Lyophilized α-syn was dissolved in 6 M GuHCl for up to 30 min to dissolve pre-formed aggregates. The solution was then loaded onto a Superdex 75 column (GE Healthcare Life Sciences, Marlborough, MA) and eluted in experimental buffer using an FPLC system (Bio-Rad, Hercules, CA). The central fraction of the eluted monomer peak was collected and the concentration was determined by absorbance at 280 nm using an extinction coefficient $\varepsilon = 5{,}800$ l mol$^{-1}$ cm$^{-1}$. The purified monomeric α-syn was always kept on ice until use in the experiments.

#### Small unilamelar vesicle preparation

Lipids were obtained lyophilized from Avanti Polar Lipids (Alabaster, AL): DOPC and DOPS. To create small unilamellar vesicles, the lipids were dissolved in chloroform:methanol 2:1 (v:v). The solvent was then evaporated in air, creating a thin lipid film deposited onto the glass vial. To ensure complete removal of the solvent, the lipid film was further dried under vacuum overnight.

The lipid films were then hydrated and dispersed by vortexing in the desired experimental buffer. For the QCM-D experiments we used an aqueous solution with a 100 mM NaCl. The lipid dispersions were sonicated using a tip sonicator for 15 minutes using a pulse sequence (10 s on/off duty and 65% amplitude) producing SUVs with typical sizes between 15 and 50 nm. The solutions were centrifuged for 5 min at 13,000 rpm and the supernatant was collected in order to remove any metal particles from the sonicator probe.

### ThT aggregation kinetics assay

To monitor the fibrillation process, 100 µl samples were aliquoted in 96-well nonbinding PEGylated plates (Half-area, 3,881 Corning plates), supplemented with 20 µM of ThT and sealed with a plastic film to avoid evaporation. Plates were incubated at 37°C in a plate reader (FluoStar Omega or FluoStar Galaxy, BMG Labtech, Offenburg, Germany) under quiescent conditions. ThT fluorescence intensity was measured in bottom reading mode with excitation filter 440 nm and emission filter 480 nm.

The experimental data were fitted by the following equation using the Amylofit online interface (Meisl et al., 2016):

$$\frac{M(t)}{M(\infty)} = 1 - \left( \frac{B_+ + C_+}{B_+ + C_+ e^{\kappa t}} \frac{B_- + C_+ e^{\kappa t}}{B_- + C_+} \right)^{\frac{k_\infty^2}{\kappa^2 ptk_\infty}} e^{-k_\infty t}, \quad (1)$$

where $B_\pm = (k_\infty \pm \check{k}_\infty)/(2\kappa)$; $C_\pm = \pm \lambda^2/(2\kappa^2)$; $\kappa = \sqrt{\{2k_+k_2m(0)^{n2+1}\}}$; $\lambda = \sqrt{\{2 \ k_+k_nm(0)^{nc}\}}$; $k_\infty = \sqrt{\{2\kappa^2/[n_2(n_2 + 1)] + 2\lambda^2/n_c\}}$ and $\check{k}_\infty = \sqrt{\{k_\infty^2 - 4C_+C_-\kappa^2\}}$.

### Preparation of α-syn seeds

α-Syn seeds were formed by adding freshly purified monomer in 10 mM MES pH 5.5 into Eppendorf tubes (Axygen low binding tubes) and left aggregating at 37°C with stirring using a teflon bar. ThT measurements of the fibrillar solution were performed to assure that the plateau was reached. For each kinetic experiment, α-syn seeds were pre-treated with 1 min continuous sonication at maximum power in a sonicator bath (Struer, Copenhagen, DK) to dispersed lumped fibrils. The concentration of seeds is counted as monomer equivalents.

### MC simulations

Peptides are modelled as flexible chains of harmonically connected beads ($r_{eq} = 7$, $k = 3k_BT/\text{Å}^2$), each representing an amino acid (Evers et al., 2012). The beads are subject to nonbonded interactions through a pair potential, $\beta u(r) = \lambda_B z_i z_j e^{-r/\lambda_D}/r + 4\beta\epsilon_{ij}\left(\left(\sigma_{ij}/r\right)^{12} - \left(\sigma_{ij}/r\right)^6\right)$ where $r$ is the amino acid separation; $z$ are charge valencies; $\lambda_B = 7$ Å the Bjerrum length in water; $\lambda_D$ is the Debye screening length; $\beta\epsilon_{ij} = 0.05$ is the Lennard–Jones (LJ) interaction strength; $\sigma_{ij}$ the LJ diameter and $\beta = 1/k_BT$ is the inverse thermal energy. The system consists of a cubic slit-geometry of side-length, $L = 500$ Å, where 208 C-terminal peptides are anchored to one side ($xy$-plane) via their N-termini. Periodic boundary conditions are applied in the $x$ and $y$ directions and a single, unconstrained α-syn chain is added. Configurational space in the Canonical ensemble ($T = 300$ K) is sampled using Metropolis MC simulations consisting of (a) single amino acid translation; (b) single peptide translation/rotation and (c) pivot rotations around randomly selected bonds. Finally (d), the protonation state of acidic and basic residues fluctuates via swap moves, associated with

an intrinsic energy cost of $(\text{pH} - \text{p}K_a) \cdot \ln 10$. A snapshot of the system is shown in Fig. 4c and the mesoscopic peptide model is described in detail elsewhere (Evers et al., 2012). All simulations were performed using the Faunus software, version 2.2 (git revision fa15d50; https://github.com/mlund/faunus) and an example input file for reproducing the setup is provided in the Supplementary Materials.

For these simulations, we estimated the area per protein C-terminal tail at both the fibril surface or the protein-coated lipid vesicle to 1,200 Å². Solid state NMR structures of α-syn fibrils reveal a density of 1 C-terminal tail per ca. 1,200 Å² fibril surface area (Li et al., 2018a; 2018b). Assuming the same saturation coverage and an average effective lipid headgroup area of 80 Å² for the mixed DOPC–DOPS bilayers at pH 6.5 and 37°C, we reach an estimate of 1,200 Å² per adsorbed protein C-terminal tail. For the present lipid system and pH, the lipid-to-protein ratio at saturation was estimated to be ca. 50/1 from kinetic experiments (Fig. S4), which is slightly higher but still within the same range as the value reported for 100% PS at pH 6.5 (Galvagnion et al., 2016). It is thus likely that the saturation point for 10% PS would be on the same order of magnitude as for 100% PS. While the membrane association of α-syn is governed by embedding of an amphiphatic α-helix in the upper acyl layer of the membrane, some negative charge – but not 100% charged lipids – is needed to accommodate the net positive charge of the membrane-bound region of the protein. The estimate of the effective lipid headgroup area at 37°C was based on literature data for DOPC and DOPS headgroups at different temperatures, which indeed show rather large variations (Lis et al. ,1982a; 1982b; Petrache et al., 2004).

**Open Peer Review.** To view the open peer review materials for this article, please visit https://doi.org/10.1017/qrd.2020.7.

**Supplementary Materials.** To view supplementary material for this article, please visit http://doi.org/10.1017/qrd.2020.7.

**Acknowledgements.** We acknowledge Håkan Wennerström and Georg Meisl for expert discussions and inputs, and Marianna Arteta, Kira Bartnik and Aleksandra Drabowska for the supplementary QCM-D experiments.

**Financial support.** This work was supported by grants from the Knut and Alice Wallenberg Foundation (ES, SL 2016.0074) and the Swedish Research Council VR (ML 2017–04372, ES 2015–00142 and SL 2015–0143). We thank LUNARC in Lund for computational resources.

**Conflict of interest.** The authors declare no conflict of interest

**Authorship contributions.** R.G., M.L., E.S. and S.L. conceived and designed the study. R.G. conducted all experiments. ML conducted all simulations. R.G., M.L., E.S. and S.L. wrote the article.

**Data availability statement.** All data will be made available through email to the authors upon reasonable request.

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
