## [Reviewer Report]

*Comments to Author*: This is a well written paper with an interesting observation about the salt dependence for synuclein aggregation with respect to secondary nucleation (using seeds) and heterogeneous assembly at surfaces (synuclein bound to vesicles with C-termini poking out). The experiments are carefully done and conditions well selected. The addition of MC simulations is good as it provides a tentative justification of the results.

The conclusion is that salt additions does not screen synuclein interactions with seeds or vesicles, although both have net negative charges, to make the interactions stronger with more salt (as one could expect from simple electrostatic considerations). Instead of net charge, it appears that local charge, ie the synuclein N-terminus that is positively charged, exhibits favorable attraction to the negative charges (of seeds and synuclein on vesicles) and this affinity dominates the interactions; thus by salt screening, interaction between synuclein monomers and synuclein seeds/ on vesicles is reduced.

In some respect, this appears as a trivial result, that the positive part of synuclein is interacting with negative parts of other synucleins on seeds or vesicles, and such interactions would be screened by addition of salt. However, it has not been shown clearly as it is here. The net charge of synuclein would be an alternative player, that is here ruled out.

The results are similar to what was found for synuclein interactions with DNA also negatively charged (reported in Kai Jiang et al Chemistry- A European Journal around 2018). Although synuclein aggregation was not probed there, and did not take place, only binding to the DNA was studied, it was shown that DNA binding of synuclein could be weakened by salt addition. This supports the current finding of local charge on synuclein (positive N terminus) being of the highest importance.

I have no critique on the experimental part and I find the results well discussed in the big picture. One question I got while reading is about the surface coverage of synuclein on the vesicles used here. PC/PS at 9:1. I would have suspected that the coverage of synuclein on such a mixture would be less than stated (as close packed as possible appears to be used in MC simulation). Other binding experiments have shown closed packed synuclein on pure PS vesicles, thus with 90 % PC, that should not bind synuclein (as pure PC vesicles do not bind synuclein), I expected a much larger footprint for synuclein.

As helpful additions for reader, to put the work in a biological context, I suggest that the authors include some mentioning of possible salt and pH ranges found in living cells and outside cells. This is relevant as addition of biological average levels of salt seems to hinder aggregation in the experiments. Still aggregation takes place in cells during disease.

Also, even if neutral pH is not the focus of the work here, it would be useful to describe the salt dependence for surface catalyzed aggregation, as well as for synuclein’s seeded and unseeded aggregation reactions in solution at neutral pH. So that is described for comparison.

---

## [Reviewer Report]

*Comments to Author*: Reviewer #1: This is a well written paper with an interesting observation about the salt dependence for synuclein aggregation with respect to secondary nucleation (using seeds) and heterogeneous assembly at surfaces (synuclein bound to vesicles with C-termini poking out). The experiments are carefully done and conditions well selected. The addition of MC simulations is good as it provides a tentative justification of the results.

The conclusion is that salt additions does not screen synuclein interactions with seeds or vesicles, although both have net negative charges, to make the interactions stronger with more salt (as one could expect from simple electrostatic considerations). Instead of net charge, it appears that local charge, ie the synuclein N-terminus that is positively charged, exhibits favorable attraction to the negative charges (of seeds and synuclein on vesicles) and this affinity dominates the interactions; thus by salt screening, interaction between synuclein monomers and synuclein seeds/ on vesicles is reduced.

In some respect, this appears as a trivial result, that the positive part of synuclein is interacting with negative parts of other synucleins on seeds or vesicles, and such interactions would be screened by addition of salt. However, it has not been shown clearly as it is here. The net charge of synuclein would be an alternative player, that is here ruled out.

The results are similar to what was found for synuclein interactions with DNA also negatively charged (reported in Kai Jiang et al Chemistry- A European Journal around 2018). Although synuclein aggregation was not probed there, and did not take place, only binding to the DNA was studied, it was shown that DNA binding of synuclein could be weakened by salt addition. This supports the current finding of local charge on synuclein (positive N terminus) being of the highest importance.

I have no critique on the experimental part and I find the results well discussed in the big picture. One question I got while reading is about the surface coverage of synuclein on the vesicles used here. PC/PS at 9:1. I would have suspected that the coverage of synuclein on such a mixture would be less than stated (as close packed as possible appears to be used in MC simulation). Other binding experiments have shown closed packed synuclein on pure PS vesicles, thus with 90 % PC, that should not bind synuclein (as pure PC vesicles do not bind synuclein), I expected a much larger footprint for synuclein.

As helpful additions for reader, to put the work in a biological context, I suggest that the authors include some mentioning of possible salt and pH ranges found in living cells and outside cells. This is relevant as addition of biological average levels of salt seems to hinder aggregation in the experiments. Still aggregation takes place in cells during disease.

Also, even if neutral pH is not the focus of the work here, it would be useful to describe the salt dependence for surface catalyzed aggregation, as well as for synuclein’s seeded and unseeded aggregation reactions in solution at neutral pH. So that is described for comparison.